# HDAC Inhibition Increases HLA Class I Expression in Uveal Melanoma

**DOI:** 10.3390/cancers12123690

**Published:** 2020-12-09

**Authors:** Zahra Souri, Aart G. Jochemsen, Mieke Versluis, Annemijn P.A. Wierenga, Fariba Nemati, Pieter A. van der Velden, Wilma G.M. Kroes, Robert M. Verdijk, Gregorius P.M. Luyten, Martine J. Jager

**Affiliations:** 1Department of Ophthalmology, LUMC, Albinusdreef 2, 2333 ZA Leiden, The Netherlands; z.souri@lumc.nl (Z.S.); m.versluis@lumc.nl (M.V.); a.p.a.wierenga@lumc.nl (A.P.A.W.); p.a.van_der_velden@lumc.nl (P.A.v.d.V.); g.p.m.luyten@lumc.nl (G.P.M.L.); 2Department of Cell and Chemical Biology, LUMC, 2333 ZA Leiden, The Netherlands; a.g.jochemsen@lumc.nl; 3Laboratory of Preclinical Investigation, Department of Translational Research, Institut Curie, PSL University, 75248 Paris, France; fariba.nemati@curie.fr; 4Department of Clinical Genetics, LUMC, 2333 ZA Leiden, The Netherlands; w.g.m.kroes@lumc.nl; 5Department of Pathology, LUMC, 2333 ZA Leiden, The Netherlands; r.m.verdijk@lumc.nl; 6Department of Pathology, Section Ophthalmic Pathology, ErasmusMC, Dr Molewaterplein 40, 3015 GD Rotterdam, The Netherlands

**Keywords:** eye diseases, uveal melanoma, oncology, HLA, HDAC, immunology, inflammation

## Abstract

**Simple Summary:**

Chemotherapy and immunotherapy are both used to treat malignancies. The immunotherapy of cancer often involves T cells, which recognise the antigens presented in HLA molecules. Uveal melanoma (UM) is an intraocular malignancy, which often gives rise to metastases. We determined whether high-risk tumours expressed the target of two drugs, histone deacetylase (HDAC) inhibitor Quisinostat and Tazemetostat, an inhibitor of Enhancer of zeste homologue 2 (EZH2). We observed that especially high-risk UM tumours (monosomy 3, gain of 8q, loss of BAP1) expressed several HDACs, and showed a high HLA Class I expression. We further tested whether these drugs influenced HLA Class I expression on three UM cell lines. The drug Quisinostat led to an upregulation of HLA protein and mRNA levels in three UM cell lines, while Tazemetostat had little effect. We concluded that the use of drugs that influence epigenetic regulators may impact immunotherapy approaches.

**Abstract:**

The treatment of uveal melanoma (UM) metastases or adjuvant treatment may imply immunological approaches or chemotherapy. It is to date unknown how epigenetic modifiers affect the expression of immunologically relevant targets, such as the HLA Class I antigens, in UM. We investigated the expression of HDACs and the histone methyl transferase EZH2 in a set of 64 UMs, using an Illumina HT12V4 array, and determined whether a histone deacetylase (HDAC) inhibitor and EZH2 inhibitor modified the expression of HLA Class I on three UM cell lines. Several HDACs (HDAC1, HDAC3, HDAC4, and HDAC8) showed an increased expression in high-risk UM, and were correlated with an increased HLA expression. HDAC11 had the opposite expression pattern. While in vitro tests showed that Tazemetostat did not influence cell growth, Quisinostat decreased cell survival. In the three tested cell lines, Quisinostat increased HLA Class I expression at the protein and mRNA level, while Tazemetostat did not have an effect on the cell surface HLA Class I levels. Combination therapy mostly followed the Quisinostat results. Our findings indicate that epigenetic drugs (in this case an HDAC inhibitor) may influence the expression of immunologically relevant cell surface molecules in UM, demonstrating that these drugs potentially influence immunotherapy.

## 1. Introduction

Uveal melanoma (UM) is a rare malignancy of the eye, estimated to occur in 6–7 cases per million per year in northern Europe and the United States of America [1,2]. The risk of developing metastases is around 50% [3]. The tumour arises from the uveal tract, which involves the choroid, ciliary body, and the iris [4]. In spite of the immunologically privileged nature of the eye, high-risk UM may show inflammation, and the presence of an inflammatory phenotype is related to a bad prognosis [5,6,7]. This inflammatory phenotype is characterised by the presence of high numbers of lymphocytes and macrophages, and a high HLA Class I and II expression [7,8,9,10]. All of these are related to the loss of one chromosome 3, a well-known risk factor for the development of metastases in this malignancy.

A high HLA Class I surface expression may protect UM cells from killing by natural killer cells (NK), and as metastases in this disease occur hematogenously, where NK cells help to remove tumour cells, a high HLA expression may help tumour cells to escape destruction [11,12,13]. On the other hand, the loss of HLA Class I expression has been identified as a tumour-escape mechanism in for instance cutaneous melanoma [14]. Both genetic and epigenetic events have been identified as regulators of HLA Class I and II expression [15], which is also the case in UM [16,17]. A study from our lab showed that one epigenetic regulator, EZH2, part of the Polycomb Repressive Complex 2 (PRC2), was found to influence HLA Class II expression in UM through the histone methylation of promoter IV of CIITA [18]. In other malignancies, members of the histone deacetylase (HDAC) family were identified as HLA regulators [19,20,21,22].

As HDACs are aberrantly expressed in UM [23], targeting these epigenetic regulators is considered for treatment [24]. In addition, the use of such inhibitors as adjuvant therapy in UM is being investigated [25,26]. A trial for the treatment of UM metastases has been set up [27], in which an immune checkpoint inhibitor (Pembrolizumab) is combined with the HDAC inhibitor Entinostat, under the hypothesis that the inhibition of HDACs may lead to an enhanced expression of HLA and cancer antigens, and a decreased activity of myeloid-derived suppressor cells. While this may help T cell-mediated immunotherapy, it may negatively affect NK cell-mediated lysis.

We set out to investigate the effect of two clinically relevant inhibitors of epigenetic modifiers, Quisinostat, an HDAC inhibitor, and Tazemetostat, an EZH2 inhibitor, on the expression of HLA Class I molecules on UM cell lines. We first determined the expression of HDACs, EZH2, as well as HLA expression in a set of 64 UMs.

## 2. Results

### 2.1. HDAC and EZH2 Expression in UM, Association with High Risk

Inhibitors of HDACs may be used as adjuvant treatment in combination with immune checkpoint inhibitors; as especially high-risk UM tumours with an inflammatory phenotype that express HLA Class I give rise to metastases, we wondered whether these high-risk tumours would express HDACs. We studied the mRNA expression of HDACs in a panel of 64 primary UMs, and compared their expression with the tumour’s chromosome 3, chromosome 8q and BAP1 status, which are indicators of a high risk of metastases formation (Figure 1A–C).

Expression of HDAC1 (*p* = 0.02), HDAC3 (*p* = 0.04), HDAC4 (*p* = 0.001), as well as HDAC8 (*p* < 0.001) was significantly higher in M3 tumours compared to D3 tumours. In contrast, HDAC11 had a significantly lower expression in high-risk M3 tumours (*p* < 0.001). With regard to both HDAC5 as well as HDAC7 expression, we found significant differences between D3 vs. M3 tumours for only one of the two probes that had been used (HDAC5 probe 1 *p* = 0.90, HDAC5 probe 2 *p* = 0.004, HDAC7 probe 1 *p* = 0.30, HDAC7 probe 2 *p* = 0.02).

The patterns for chromosome 8q gain (Figure 1B) and BAP1 expression (Figure 1C) resembled those of chromosome 3. HDAC1 (*p* = 0.05), HDAC3 (*p* = 0.02), the second probe of HDAC5 (*p* = 0.04) and of HDAC7 (*p* = 0.03), and HDAC8 (*p* = 0.004), were increased in tumours with 8q gain, while HDAC11 was decreased (*p* = 0.002). The loss of BAP1 expression as assessed by immunohistology was associated with an increase in HDAC4 (*p* = 0.003), the second probe of HDAC5 (*p* = 0.03) and HDAC7 (*p* = 0.02) and a decrease in HDAC11 (*p* < 0.001).

We also looked at EZH2, which is known to be higher in UM with a high mitotic count [28]. However, the EZH2 expression was not correlated with chromosome 3 or 8q status, or BAP1 status (*p* = 0.24, *p* = 0.06, *p* = 0.24).

### 2.2. Association between HDAC and EZH2 Expression and HLA Class I in UM

As high-risk tumours are known to have a higher HLA Class I expression than low risk tumours, we hypothesised that the expression of HLA Class I and M3-associated HDACs would be correlated. We tested this for all HDACs and EZH2 in our set of 64 tumours.

Expression of HDAC1, HDAC4, and HDAC8 was convincingly positively correlated with HLA-A and/or HLA-B (Table 1). The two HDAC7 probes showed diverging results, with the M3-associated probe being positively correlated with one HLA-A probe and with the HLA-B probe. The HDAC11 expression showed a significant negative correlation with HLA-A and -B expression (all *P*s < 0.001). These data indicate that the expression levels of three of the four HDACs, which are upregulated in M3 tumours, correlate with a high HLA-A and -B expression. HDAC11 shows the opposite pattern. No significant correlation was seen between the EZH2 and HLA Class I expression in this set of 64 tumours.

### 2.3. In Vitro Analysis of Effect of HDAC and EZH2 Inhibition on UM Cell Lines

As mentioned earlier, high-risk M3 UMs contain more infiltrating cells, which also express HLA antigens. The correlation between the expression of certain genes and high-risk tumours is, therefore, difficult to interpret, as it cannot be excluded that the increased levels of, e.g., HLA Class I and certain HDACs, is caused to a significant extent by the increased number of infiltrating cells and not so much by the UM tumour cells themselves.

To be able to investigate an effect of HDAC activity on HLA Class I expression, we made use of the pan-HDAC inhibitor Quisinostat and treated the UM cell lines in vitro.

Recent studies have shown that HDAC inhibitors are able to reduce growth in UM cell lines [29,30], but to our knowledge, no one has reported on the effect of epigenetic enzyme inhibition on the expression of immune modulators, such as HLA Class I, in UMs. We also used an EZH2 inhibitor, Tazemetostat, and studied the effect of combination therapy.

#### 2.3.1. Morphological Analysis of UM Cell Lines after 48 h Treatment with Quisinostat and Tazemetostat

First of all, we looked at the morphological characteristics of the cell lines after 48 h treatment with Quisinostat and Tazemetostat compared with untreated cell lines (Figure 2). In the presence of Quisinostat (Q, 40 nM), at 48 h, the cell sizes of all three cell lines was reduced and more floating cells were visible. Upon treatment with Tazemetostat (T, 5 µM), at 48 h, no large differences in the morphology or confluency were observed compared to the controls. Cells which were treated with a combination of the two drugs (40 nM Q + 5µM T) changed morphologically and resembled Quisinostat single treatment: plates with OMM2.5 and MP38 seemed to have fewer cells than when treated with Quisinostat alone.

#### 2.3.2. Quisinostat and Cell Growth

A quantitative analysis of cell numbers after 24 and 48 h using the Cell Titer-Blue Cell Viability Assay (Figure 3) confirmed these findings: Quisinostat reduced the cell numbers of OMM2.5 and MP38 cells after 24 h (*p* = 0.01, *p* = 0.03), and of all three cell lines after 48 h (*p* < 0.001, *p* = 0.002, *p* < 0.001). Tazemetostat by itself did not influence cell numbers. However, at 48 h, the combination of both drugs (QT) decreased cell numbers significantly in two cell lines (OMM1 and MP38) compared to the Q treatment alone (*p* < 0.001 for both).

#### 2.3.3. Effect of Quisinostat and Tazemetostat on HLA Class I Cell Surface Expression

We investigated the effect of Quisinostat and Tazemetostat on HLA Class I surface expression of the UM cell lines by flow cytometry after 48 h treatment. The three cell lines (OMM1, OMM2.5, and MP38) expressed HLA Class I, as determined using monoclonal antibody W6/32 (Figure 4A–C, controls). Quisinostat increased HLA Class I surface expression in all three cell lines (*p* < 0.01). Tazemetostat, on the other hand, did not affect the HLA Class I surface expression in the cell lines, while the combination treatment resulted in a significant increase compared to Quisinostat alone in OMM2.5 and MP38 (*p* = 0.003 and *p* = 0.004).

#### 2.3.4. Effect of Quisinostat and Tazemetostat on HLA Class I mRNA Expression by qPCR

We investigated the level of HLA Class I mRNA by qPCR after 48 h of treatment with Quisinostat and/or Tazemetostat. HLA-A and HLA-B expressions were determined separately (Figure 4D–F and Figure 4G–I).

In agreement with what we found regarding broad HLA Class I cell surface expression, Quisinostat increased HLA-A in all three cell lines (*p* < 0.01). Tazemetostat gave a slight increase in cell line OMM2.5, while in all three cell lines, combined treatment led to a lower expression than with Quisinostat alone. With regard to HLA-B, Quisinostat led to a higher expression in all three cell lines, while Tazemetostat by itself gave a significant but slight increase in MP38. The combination of the two drugs led to a high expression on all three cell lines, with divergent results when compared to Quisinostat alone.

## 3. Discussion

HDACs are a group of epigenetic regulators which have been shown to regulate inflammation in some malignancies [31,32]. Moreover, elevated HDAC expression has been associated with other inflammation-related diseases such as chronic hepatitis B [33] and rheumatoid arthritis (RA) [34]. To our knowledge, the relationship between the expression of these epigenetic regulators and the inflammatory phenotype in UM has not been investigated. As combinations of chemotherapy and immunotherapy are being used as adjuvant treatments and to treat metastases in e.g., cutaneous melanoma, we investigated the expression of several epigenetic regulators and their potential involvement in the regulation of HLA Class I, a molecule which plays an important role in immune responses, in UM.

In a set of 64 UMs, the expression of HDACs 1, 4, and 8 was positively correlated with the expression of HLA Class I. Using malignant cell lines such as from colorectal carcinoma, e.g., HDAC1 has been shown to be related to inflammation, by inducing STAT1 activation and the transcription of downstream targets of the IFN-Ƴ signalling pathway [35,36]; such targets include HLA Class I components. In addition, we show that particularly HDAC1, 3, 4, and 8 are expressed at higher levels in high-risk UM.

We tested the effects of two drugs, Quisinostat and Tazemetostat. The pan-HDAC inhibitor Quisinostat has been shown to effectively reduce UM cell viability in vitro [29,37], and is currently being tested in clinical studies of advanced solid tumours and leukaemia [38]. Growth inhibition was accomplished by the induction of cell cycle arrest and apoptosis in other cancers such as lung cancer [39], and hepatocellular carcinoma [40]. Tazemetostat is a drug known to inhibit EZH2 and has been shown to reduce growth in medulloblastoma [41], B-cell non-Hodgkin lymphoma and epithelioid sarcoma [42]. When we applied the pan-HDAC inhibitor Quisinostat to UM cell lines, HLA Class I cell surface and mRNA expression increased, while the effect of Tazemetostat was marginal. As Quisinostat is a pan-HDAC inhibitor [30], the effect is likely due to the inhibition of deacetylation. Indeed, we have previously shown that treatment with Quisinostat strongly increased the level of acetylated Histone 3 [29]. In general, higher acetylation coincides with less chromatin compactness, higher promoter activity and increased gene transcription. We have no evidence that the effect of HDAC inhibition on HLA Class I gene transcription is caused by a direct effect on the chromatin modification around the Class I promoters. However, our results are generally in agreement with studies in other malignancies describing that HDAC inhibition will upregulate HLA Class I and some HLA regulatory components [19,20,21,22,43]. Whether increasing HLA expression in UM is helping the patient may depend on the situation: it may help to stimulate T cell-mediated cytotoxicity against metastases; however, when the primary UM is still in place, increasing HLA expression may limit the NK cell activity directed against metastatic cells which spread hematogeneously.

HDAC11 expression was significantly lower in high-risk tumours (Figure 1A–C). In 2014, Herlihy and colleagues already showed a low expression of HDAC11 in a small series of M3 UMs [23]. The gene of this HDAC is located on chromosome 3, and the loss of chromosome 3 in high-risk UMs most likely explains these findings. However, it does not necessarily mean that low HDAC11 expression has no causal relationship with an increased risk of the development of metastases. Although most studies on the function of HDAC11 suggest an oncogenic role, as in breast cancer the downregulation of HDAC11 provided the cells with an increased ability to invade from the lymph nodes to other organs [44]. It is therefore possible that the low HDAC11 expression may help the hematogenous spread to the liver.

In UMs, EZH2 plays a major role in silencing promoters and inhibiting transcription [18] and is associated with a high mitotic count [28]. Although this enzyme represses CIITA transcription by the K27m3 of histone 3, thereby silencing HLA Class II expression, we observed a high expression of EZH2 in high-risk M3 tumours, without any association to HLA Class I expression. EZH2 inhibition with Tazemetostat did not have an effect on the cell surface expression of HLA Class I, while we observed a slight increase in HLA-A mRNA in OMM2.5 and HLA-B mRNA in cell line MP38. The combination with Quisinostat led to results that varied between cell lines and HLA allele.

A Phase 1 trial is planned to use the HDAC 1, 2, and 3 inhibitor Domatinostat together with immune checkpoint inhibitors as the neo-adjuvant treatment of cutaneous melanoma patients with lymph node metastases (ClinicalTrials.gov NCT-4133048). This trial is set up under the assumption that the HDAC inhibitor will stimulate the interferon pathway, making the tumour “hot”. Such tumours may contribute to a good induction of cytotoxic T cells. As already mentioned earlier, high-risk UMs already have a high HLA expression, and it is unclear whether the same approach will help to treat the metastases of UMs. While HLA expression is important for tumour cell killing by T cells, a high expression blocks recognition by NK cells. Furthermore, the expression of other cell surface molecules such as checkpoint inhibitors may also be modified by blocking epigenetic regulators.

## 4. Materials and Methods

### 4.1. Study Population

Tumours were derived from 64 eyes that underwent an enucleation for UMs between 1999 and 2008 at the Leiden University Medical Center (LUMC), Leiden, The Netherlands; 51% of the patients were male and 49% female. Their mean age at the time of enucleation was 61 years.

This work has been carried out in accordance with the Code of Ethics of the World Medical Association (Declaration of Helsinki). The project was approved by the LUMC Biobank committee and the LUMC METC committee (19 October 2016, code G16.076/NV/gk).

### 4.2. Chromosome Analysis

DNA was isolated using the QIAmp DNA Mini kit (Qiagen, Venlo, The Netherlands). Single-nucleotide polymorphism (SNP) analysis was performed using an Affymetrix 250K_NSP or Affymetrix SNP 6.0 array to detect chromosome 3 and 8 abnormalities [45].

### 4.3. Immunohistochemistry

Tissues were incubated with mouse monoclonal antibody (clone sc-28383, 1:50 dilution, Santa Cruz Biotechnology, Dallas, TX, USA) for BAP1 evaluation [46,47]. Nuclear staining was evaluated by an experienced ocular pathologist.

### 4.4. Tumour Gene Expression

Gene expression profiling was performed with the Illumina HT12v4 array (Illumina, Inc., San Diego, CA, USA) and the data were obtained for the expression of epigenetic regulators (HDAC1, HDAC2, HDAC3, HDAC4, HDAC5, HDAC6, HDAC7, HDAC8, HDAC9, HDAC11, EZH2) and HLA Class I genes (HLA-A, HLA-B) as described previously [16]. Information regarding the Illumina probe numbers and gene expression levels is shown in Appendix A.

### 4.5. Reagents

Quisinostat and Tazemetostat were purchased from Selleckchem (Houston, TX, USA). Drugs were dissolved in dimethyl sulfoxide (DMSO) to reach a stock concentration of 5 mM and diluted in the indicated fresh medium for in vitro studies.

### 4.6. Cell Lines and Cell Culture

The OMM1 cell line was established by Dr G.P.M. Luyten (LUMC, Leiden, The Netherlands) [48], while OMM2.5 was a gift from Dr B.R. Ksander (Schepens Eye Research Institute, Boston, MA, USA) [49]. Both were cultured in Roswell Park Memorial Institute Medium 1640 (RPMI) Dutch modified media (Life technologies, Europe bv, Bleiswijk, The Netherlands) supplemented with 10% foetal bovine serum (FBS) (Greiner Bio-one, Alphen aan den Rijn, The Netherlands), 1% GlutaMAX and 1% penicillin/streptomycin (Life technologies). MP38 was kindly provided by the Curie Institute, Paris, France [50] and cultured in Iscove’s modified Dulbecco medium (IMDM) (Life Technologies) containing 20% FBS (Greiner Bio-one) and 1% penicillin/streptomycin (Gibco). Cells were incubated in 5% CO2 at 37 °C in monolayers in tissue culture flasks in a humidified incubator.

### 4.7. Cell Proliferation Study

Cells were seeded in triplicate in 135 µL in 96-well plates; 24 h after seeding the cells, 15 µL of 10× concentrated drug in media was added to each well. Cell viability was determined by the Cell Titer-Blue Viability Assay: after removing the medium, 100 µL of appropriately diluted CellTiter-Blue Reagent (Promega, Fitchburg, WI, USA) was added to each well and after 60 min, fluorescence was measured in a microplate reader (Victor X3, Perkin Elmer, San Jose, CA, USA).

### 4.8. Flow Cytometry

Cells were incubated with an optimal dilution of mouse monoclonal anti-HLA Class I antibody (W6/32, 311414, Alexa Fluor 647; Bio Legend, Amsterdam, The Netherlands) and the data were collected (10,000–15,000 cells per live gate) using the BD LSR II system (BD Biosciences, San Diego, CA, USA); results were analysed using FACSDiva software (BD Biosciences). Three independent experiments have been performed.

### 4.9. RNA Isolation and Quantitative Real-Time Polymerase Chain Reaction (qPCR)

RNA isolation and quantitative real-time polymerase chain reaction (qPCR) have been described previously [51]. Briefly, total RNA was extracted from cell cultures using an RNeasy Mini Kit (Qiagen Benelux B.V., Venlo, The Netherlands). Using the iScript cDNA synthesis kit (Bio-Rad Laboratories B.V., Veenendaal, The Netherlands), complementary cDNA was synthesised. Quantitative real-time polymerase chain reaction (qPCR) was performed in three independent experiments using the CFX-384 machine, with triplicates of one experiment used to make the graphs. CFX manager 3.1 (Bio-Rad) software was used to analyse the data, and the CT values of genes of interest were normalised against the geometric mean of housekeeping genes *RPS11*, *CAPNS1*, and *SRPR*. The sequences of primers used in the study are shown in Appendix A.

### 4.10. Statistical Analysis

Data were analysed with SPSS software version 22.0 (SPSS, nc. Chicago, IL, USA). Spearman correlation was performed in order to test correlations between non-parametric data. Bonferroni correction was applied for multiple testing, where appropriate. The Mann–Whitney U test was used to compare non-normal groups. Graphs were obtained using GraphPad Prism version 5.0 for Windows (GraphPad Software, La Jolla, CA, USA). An independent t-test was used to compare the difference between the means of the in vitro studies.

## 5. Conclusions

To the best of our knowledge, this is the first study which reports the association between HDAC epigenetic regulators and HLA Class I expression in UMs. We showed that high-risk UMs not only show a higher expression of HLA Class I antigens but also of several HDACs. The pan-HDAC inhibitor Quisinostat increases HLA Class I expression at the protein and mRNA level while EZH2 inhibition by Tazemetostat has a slight effect on HLA expression. As epigenetic anti-cancer drugs may influence the expression of immunologically relevant molecules such as HLA in UMs, they may influence the function of T cells and NK cells, and thereby the effect of immunotherapy.

## Figures and Tables

**Figure 1 cancers-12-03690-f001:**
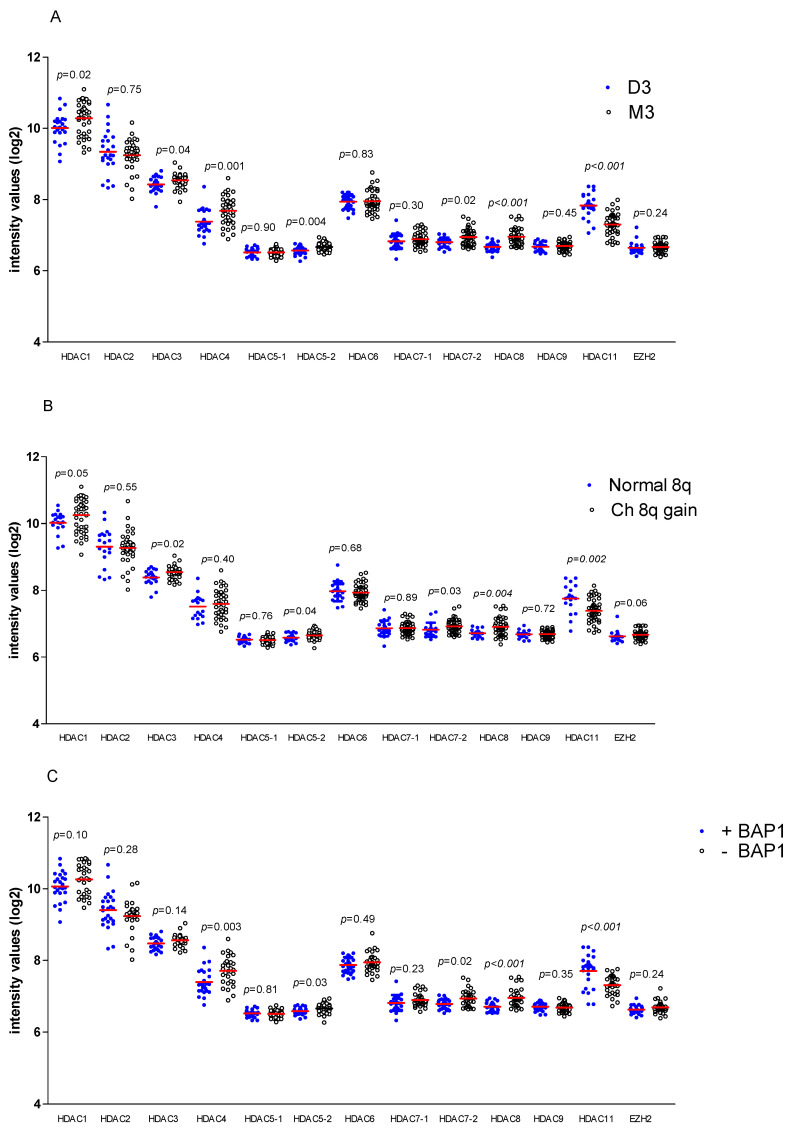
The expression of histone deacetylases (HDACs) in uveal melanoma in relation to the tumour’s chromosome 3, chromosome 8q and BAP1 status. The expression of several HDACs and EZH2 was compared between tumours with (**A**) disomy for chromosome 3 (D3, *n* = 24) and a monosomy of chromosome 3 (M3, *n* = 40); (**B**) normal 8q (*n* = 19) vs. 8q gain (*n* = 45); (**C**) positive staining for BAP1 (+BAP1, *n* = 25) vs. negative staining for BAP1 (-BAP1, *n* = 30). A Mann–Whitney U test was applied. Horizontal bars indicate mean gene expression.

**Figure 2 cancers-12-03690-f002:**
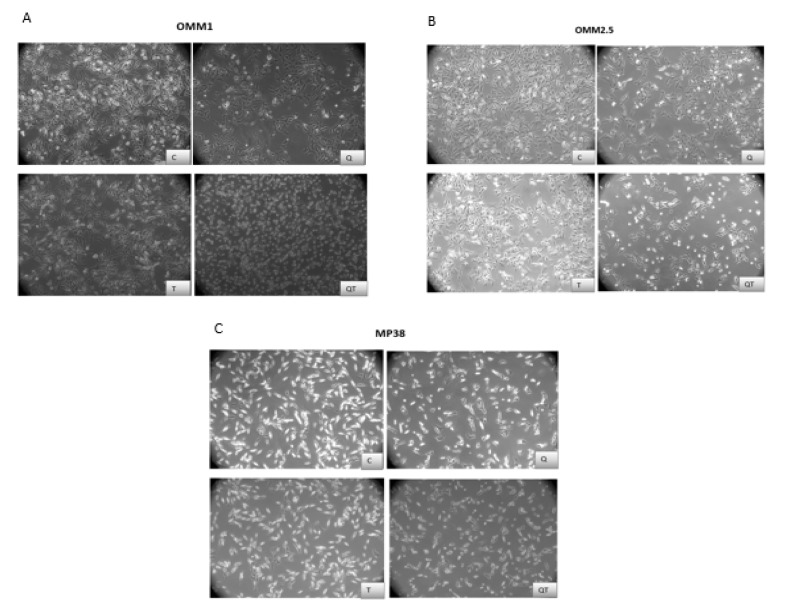
Photographs of uveal melanoma cell lines taken after 48 h exposure to Quisinostat (Q) and/or Tazemetostat (T): the photographs (magnification 100×) show fewer and more rounded cells after incubation with 40 nM Q and after combination treatment. (**A**) = cell line OMM1, (**B**) = cell line OMM2.5, (**C**) = cell line MP38. C: control; Q: 40 nM Quisinostat; T: 5 µM Tazemetostat; QT: combination of 40 nM Quisinostat and 5 µM Tazemetostat.

**Figure 3 cancers-12-03690-f003:**
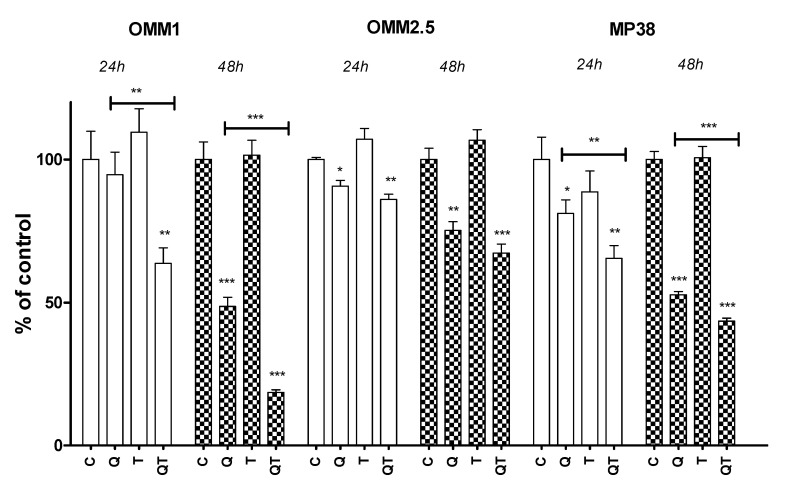
Cell numbers of three uveal melanoma cell lines after 48 h of treatment with Quisinostat or Tazemetostat of uveal melanoma cell lines. Cell density was determined using a cell titer blue assay. C: control; Q: 40 nM Quisinostat; T: 5 µM Tazemetostat; QT: combination of 40 nM Quisinostat and 5 µM Tazemetostat. * *p* ≤ 0.5, ** *p* < 0.01, *** *p* < 0.001. Error bars indicate the standard error of the mean.

**Figure 4 cancers-12-03690-f004:**
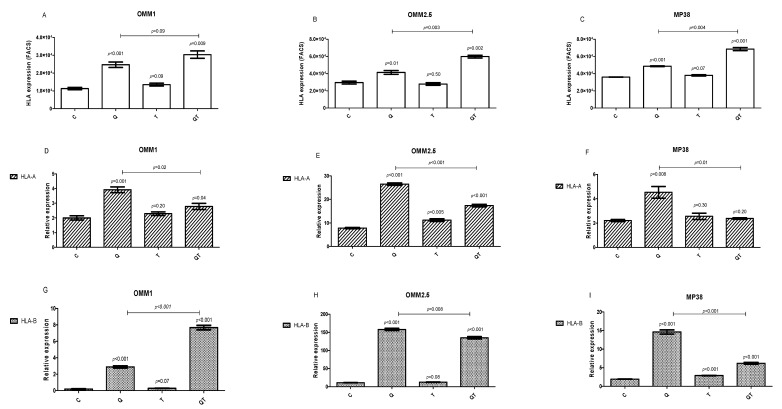
HLA Class I cell surface protein expression and mRNA levels after 48 h of treatment with Quisinostat and/or Tazemetostat compared to the control. Cell surface expression was determined by FACS using anti-HLA W6/32 antibody (**A**–**C**). HLA-A mRNA (**D**–**F**) and HLA-B mRNA (**G**–**I**) expression was determined by qPCR. C: control; Q: 40 nM Quisinostat; T: 5 µM Tazemetostat; QT: combination of 40 nM Quisinostat and 5 µM Tazemetostat. Using an Independent t-test, *p* ≤ 0.05 was considered significant. Error bars indicate the standard error of the mean of three measurements.

**Table 1 cancers-12-03690-t001:** Correlation between mRNA expression levels (determined by Illumina array) of different HDACs and EZH2 and the expression of HLA-A (three probes) and HLA-B. R = two-tailed Spearman correlation coefficient. Following Bonferroni correction, *p* ≤ 0.005 is considered significant, and indicated in bold.

	HLA-A Probe 1		HLA-A Probe 2		HLA-A Probe 3		HLA-B	
	R	*p*	R	*p*	R	*p*	R	*p*
**HDAC1**	0.367	**0.003**	0.392	**0.001**	0.380	**0.002**	0.370	**0.003**
**HDAC2**	−0.144	0.26	−0.167	0.19	−0.108	0.40	−0.162	0.20
**HDAC3**	0.262	0.04	0.241	0.05	0.293	0.02	0.318	0.01
**HDAC4**	0.143	0.26	0.371	**0.003**	0.354	**0.004**	0.305	0.01
**HDAC5 probe 1**	0.029	0.82	0.090	0.48	0.018	0.88	−0.045	0.72
**HDAC5 probe 2**	0.091	0.47	0.231	0.07	0.096	0.45	0.151	0.23
**HDAC6**	−0.179	0.16	−0.136	0.28	−0.284	0.02	−0.196	0.12
**HDAC7 probe 1**	−0.020	0.87	0.010	0.94	−0.014	0.91	0.108	0.39
**HDAC7 probe 2**	0.225	0.07	0.382	**0.002**	0.263	0.04	0.352	**0.004**
**HDAC8**	0.456	**<0.001**	0.588	**<0.001**	0.539	**<0.001**	0.550	**<0.001**
**HDAC9**	−0.066	0.60	−0.091	0.48	−0.209	0.01	−0.190	0.13
**HDAC11**	−0.541	**<0.001**	−0.548	**<0.001**	−0.515	**<0.001**	−0.607	**<0.001**
**EZH2**	0.211	0.09	0.214	0.09	0.226	0.07	0.146	0.25

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
