# Peer review of "HDAC Inhibition Increases HLA Class I Expression in Uveal Melanoma"

_cancers, 2020, doi:10.3390/cancers12123690_

Round 1

Reviewer 1 Report

I would like to congratulate the authors on the manuscript entitled "HDAC inhibition increases HLA Class I expression in Uveal Melanoma". The article is well written, references are appropriated.

Few comments/questions:

The paragraph "To the best of our knowledge this is the first study which reports the association between HDAC epigenetic regulators and the inflammatory phenotype in UM" should be changed. It can create confusion to the readers since there are already publications regarding HDAC inhibitors for UM (two examples below).

HDAC Inhibition Enhances the In Vivo Efficacy of MEK Inhibitor Therapy in Uveal Melanoma. Faião-Flores F, Emmons MF, Durante MA, Kinose F, Saha B, Fang B, Koomen JM, Chellappan SP, Maria-Engler SS, Rix U, Licht JD, Harbour JW, Smalley KSM.Clin Cancer Res. 2019 Sep 15;25(18):5686-5701. doi: 10.1158/1078-0432.CCR-18-3382. Epub 2019 Jun 21.   In vitro and in vivo anti-uveal melanoma activity of JSL-1, a novel HDAC inhibitor. Wang Y, Liu M, Jin Y, Jiang S, Pan J.Cancer Lett. 2017 Aug 1;400:47-60. doi: 10.1016/j.canlet.2017.04.028. Epub 2017 Apr 26.PMID: 28455241   The authors only used monosomy 3 as a marker for high risk uveal melanoma and no comments are made on other chromosomes (1, 6, 8), gene expression profile, BAP1, GNAQ, GNA11 and others.

Author Response

We thank the reviewer for his/her kind remarks.

Comment 1: The paragraph "To the best of our knowledge this is the first study which reports the association between HDAC epigenetic regulators and the inflammatory phenotype in UM" should be changed. It can create confusion to the readers since there are already publications regarding HDAC inhibitors for UM

Our answer: We modified our sentence in line 313 to: “To the best of our knowledge this is the first study which reports the association between HDAC epigenetic regulators and HLA Class I expression in UM.”

Comment 2: The authors only used monosomy 3 as a marker for high risk uveal melanoma and no comments are made on other chromosomes (1, 6, 8), gene expression profile, BAP1, GNAQ, GNA11 and others.

Our answer: We added two more figures with information on the relation between HDAC expression and Chromosome 8q and BAP1 staining, as these are indeed also high risk factors in UM; because of the added information on this, we adapted sentences in the simple summary, in part 2.1 of the results, and modified the discussion.

In addition, we added information in the materials and methods section and therefore modified line 289, added part 4.3 (information on BAP1 staining) and added two references for BAP1 staining.

As we added information on chromosome 8q, we invited the researcher who provided us with the chromosome information, Dr Kroes. Dr Kroes accepted co-authorship.

Reviewer 2 Report

The paper is focused on the investigation the effect of two clinically-relevant inhibitors of epigenetic modifiers,  Quisinostat, an HDAC inhibitor, and Tazemetostat, an EZH2 inhibitor, on the expression of HLA Class I molecules on UM cell lines.
Results obtained in this study could change a treatment approach and help UM patients with metastases.
The methods, research tools and statistical analysis are clearly defined and well explained.
The authors have presented a large knowledge of discussed issues and cited references.

The paper needs a few, minor improvements.
Minor comments:
1. Figure 1. Horizontal bars indicate mean gene expression for black spot M3 should be replace for different colour than black, to better visualisation.

2. Figure 4. I couldn't find error bars indicate the standard error of the mean of three measurements in A, B, C, D, E and F graphs.

Author Response

Reviewer 2:

We thank the reviewer for her/his kind words.

Comment 1: Figure 1. Horizontal bars indicate mean gene expression for black spot M3 should be replaced for different color than black, to better visualization.

Our answer: We changed the horizontal bar color in figure 1 (which indicate mean of expression) from black to red for better visualization.

Comment 2: Figure 4. I couldn't find error bars indicate the standard error of the mean of three measurements in A, B, C, D, E and F graphs.

Our answer: Bars indicating standard error of the mean are sometimes so small that they cannot be seen. We made them thicker for better visualization.